# Long-Term Saccharin Consumption and Increased Risk of Obesity, Diabetes, Hepatic Dysfunction, and Renal Impairment in Rats

**DOI:** 10.3390/medicina55100681

**Published:** 2019-10-09

**Authors:** Omar Hasan Azeez, Suad Yousif Alkass, Daniele Suzete Persike

**Affiliations:** 1Department of Pathology and Microbiology, College of Veterinary Medicine, University of Dohuk, Duhok 1006 AJ, Iraq; 2Department of Medicinal Chemistry, College of Pharmacy, University of Duhok, Duhok 1006 AJ, Iraq; suad.alkass@uod.ac (S.Y.A.); daniele_persike@protonmail.com (D.S.P.)

**Keywords:** Sodium saccharin, oxidative stress, hyperglycemia, obesity, kidney and liver disfunction

## Abstract

*Background and objectives:* This study evaluated the effect of chronic consumption of saccharin on important physiological and biochemical parameters in rats. *Materials and Methods:* Male Wistar rats were used in this study and were divided into four groups: A control group and three experimental groups (groups 1, 2, and 3) were treated with different doses of saccharin at 2.5, 5, and 10 mg/kg, respectively. Each experimental group received sodium saccharin once per day for 120 days while the control group was treated with distilled water only. In addition to the evaluation of body weight, blood samples [total protein, albumin, glucose, lipid profile, alanine transaminase (ALT), aspartate transaminase (AST), lactate dehydrogenase (LDH), creatinine, and uric acid] and urine (isoprostane) were collected in zero time, and after 60 and 120 days for biochemical evaluation. Liver (catalase activity) and brain (8-hydroxy-2’-deoxyguanosine, 8-OHdG) tissues were collected at time zero and after 120 days. *Results:* The data showed that saccharin at 5 mg/kg increased body weight of treated rats after 60 (59%) and 120 (67%) days of treatment. Increased concentration of serum glucose was observed after treatment with saccharin at 5 (75% and 62%) and 10 mg/kg (43% and 40%) following 60 and 120 days, respectively. The concentration of albumin decreased after treatment with saccharin at 2.5 (34% and 36%), 5 (39% and 34%), and 10 mg/kg (15% and 21%) after 60 and 120 days of treatment, respectively. The activity of LDH and uric acid increased proportionally with dosage levels and consumption period. There was an increased concentration of creatinine after treatment with saccharin at 2.5 (125% and 68%), 5 (114% and 45%), and 10 mg/kg (26% and 31%) following 60 and 120 days, respectively. Catalase activity and 8-OHdG increased by 51% and 49%, respectively, following 120 days of treatment with saccharin at 2.5 mg/kg. Elevation in the concentration of isoprostane was observed after treatment with saccharin at all doses. *Conclusions:* The administration of saccharin throughout the treatment period was correlated with impaired kidney and liver function. Both hyperglycemic and obesity-inducing side effects were observed. There was an increased oxidative status of the liver, as well as exposure to increased oxidative stress demonstrated through the increased levels of isoprostane, uric acid, 8-OHdG, and activity of catalase. Therefore, it is suggested that saccharin is unsafe to be included in the diet.

## 1. Introduction

The use of non-nutritive artificial sweeteners (NAS) in place of nutritive sweeteners such as sugar is recommended as a way of reducing overall carbohydrate and calorie intake [1]. Patients with diabetes and obese/overweight individuals often use NAS as a tool for glycemic control and weight loss. Health-conscious consumers are more likely to use artificial sweeteners to reduce their risk of weight gain and metabolic disease [2]. Aside from medical treatment, diets with limited quantities of sugar and carbohydrate-rich foods are considered to be the most effective means of managing body weight [1]. 

### 1.1. Saccharin Chemical Characteristics and Uses

Saccharin (1,2-benzisothiazol-3(2H)-one-1,1-dioxide) (see Figure 1) is the oldest artificial sweetener, and was discovered in 1879 [3]. The compound is prepared through reacting methyl anthranilate with nitrous acid sulfur dioxide, chlorine, and ammonia [4]. It is about 300 times sweeter than sucrose and is considered to be one of the most important and widely used sweeteners worldwide [5].

Saccharin is a water-soluble acid with a pKa of 1.8. Its absorption is increased in animal species with lower stomach pH, such as rabbits and humans, relative to other mammals with higher stomach pHs such as rats [6,7,8]. Other forms of saccharin that are consumed include: calcium saccharin, potassium saccharin, and acid saccharin. Sodium saccharin is used most often due to its greater palatability [9]. 

Saccharin, in addition to being used as a table-top sweetener, is commonly used in soft drinks, baked foods, jams, canned fruit, candy, dessert toppings, and chewing gum [10]. Since saccharin’s sweetening power is not reduced when heated, it is an excellent candidate as an additive in low-calorie and sugar-free products [11,12]. 

### 1.2. Pharmacokinetic Characteristics of Saccharin

Unlike glucose and sucrose, saccharin cannot be metabolized by the body [11]. Saccharin has been reported as a stable compound under a wide range of conditions and no detectable metabolism of saccharin was shown either in animals or humans [6]. Current research hypothesizes that because saccharin is highly polar, it is slowly and incompletely absorbed from the gut, but rapidly eliminated in the urine, leading to a reduction in its concentration within the plasma. The decrease in plasma concentrations after oral dosing is governed by the absorption rate, which is a phenomenon described as a flip-flop situation typical of molecules which are absorbed slowly but eliminated rapidly [13]. 

Since saccharin is not metabolized and does not produce food energy, it has become an important sweetener, especially for diabetics [14]. The affect of saccharin on insulin release is controversial. Ionescu et al. (1988) reported that saccharin triggered the release of insulin in genetically modified obese rats due to its taste [14], whereas Whitehouse et al. (2008) showed that saccharin did not affect blood insulin levels in animals that were treated with the sweetener from the time of conception to death [15].

### 1.3. Regulations Concerning Saccharin

The Food and Drug Administration (FDA) and the Federal Agency of the United States Department of Health and Human Services (DHHS) consider saccharin to be a safe sweetener [16].

The Joint Food and Agriculture Organization/World Health Organization Expert Committee on Food Additives (JECFA), European Union, US FDA, Japan, France, China, and Taiwan established an acceptable daily intake (ADI) of saccharin at 5 mg/kg body weight [17].

### 1.4. Saccharin and Toxicity

Saccharin and toxicity are arguable. Throughout the 1960s, various studies suggested that saccharin might be an animal carcinogen [18]. Yilmaz and Uçar (2015) stressed that genotoxicity and carcinogenicity of saccharin are not understood clearly [12]. Most publications reference that saccharin increases the rate of bladder cancer in rats fed with large doses [19,20,21,22,23]. 

A few epidemiological studies also found relationships between saccharin and bladder cancer risk in humans [17,18,19,20], but the majority of studies found no association between saccharin and cancer [21,22,23]. Sodium saccharin has shown tumorigenic effects in rats. The compound was reported to produce hyperplastic response within a relatively short period of time when administered at high doses (≥2.5%) [24].

Saccharin consumption has been associated with adverse effects on most of the biochemical and hematological blood indices in rats [4]. Chronic saccharin intake affects biochemical parameters, and reported findings reflect various metabolic, hormonal, and neural responses in male and female rats resulting from the prolonged use of this sweetener after a single dose in drinking water [25]. Consumption of large amounts of saccharin (135 mg) may result in hypoglycemia [26], reduced hyperinsulinemia, decreased insulin resistance, and improved glycemic control in hyperglycemic obese mice [27]. 

### 1.5. Saccharin and Oxidative Stress

Saccharin may induce oxidative stress on the liver cells through lowering catalase activity and the total antioxidant concentration (TAC) in plasma [28]. It was demonstrated that saccharin harmfully affects both hepatic and renal tissues and alters biochemical markers, not only at high doses, but also at low doses in rats [29].

Different methodologies have been applied to assess oxidative stress in animal models and humans. DNA bases are highly susceptible to reactive oxygen species (ROS) oxidation. The predominant detectable DNA oxidation product in vivo is 8-hydroxy-2′-deoxyguanosine (8-OHdG). 8-OHdG was established as sensitive marker of DNA damage, becoming a very important biomarker for processes such us carcinogenesis, aging, and degenerative diseases [30,31].

During the process of lipid peroxidation, a number of compounds are formed including alkanes, malondialdehyde, and isoprostanes, which are used as markers in lipid peroxidation assay [32]. The discovery of isoprostanes has important implications for medicine as described by Morrow and Roberts [33]. Measurement of F2-isoprostane is the most reliable approach to assess oxidative stress, providing an important tool to explore the role of oxidative stress in the pathogenesis of human disease [34]. 

Due to the increasing demand for the use of non-nutritive artificial sweeteners as a tool to control the level of blood glucose and body weight, and due to uncertain and little information on the safety of using saccharin as a sweetener, the present study was set to analyze the effect of long-term consumption of saccharin on biochemical parameters when increasingly large doses are administered.

## 2. Materials and Methods

### 2.1. Animals

60 male Wistar rats (3–4 months), weighing 250–325 g, were housed in the animal house of the College of Veterinary Medicine - University of Duhok. These animals were kept in ventilated cages at a controlled temperature (22 ± 2°C) and were exposed to cycles of light and dark. Food and water were given ad libitum. Rat handling and treatment were according to guidelines for laboratory animal care and use [35]. The study was approved by the Animal Ethics Committee of the College of Veterinary Medicine - University of Duhok (Ethical code No. DR1996919CV, approved on the 11th of June, 2019).

### 2.2. Biochemical Assays

#### 2.2.1. Chemicals

Sodium saccharin was purchased from Alfa Aesar Thermo Fisher Scientific, Germany. Colorimetric assay kits from Biolabo (France) were used to measure the concentration of glucose, total cholesterol (TC), high density lipoprotein (HDL-cholesterol; HDL), low density lipoprotein (LDL-cholesterol; LDL), triglycerides (TG), total protein (TP), serum albumin, uric acid, creatinine, activity of the enzymes lactate dehydrogenase (LDH), aspartate transaminase (AST), and alanine aminotransaminase (ALT). The activity of serum catalase was assayed according to Hadwan and Abeds (2016) [36]. Urinary Isoprostane and 8-OHdG were purchased from MyBioSource (San Diego CA, USA).

#### 2.2.2. Experimental Design

Rats were randomly distributed into four groups, with 15 rats in each group. For the experiment baseline (time zero), five rats from each group were euthanized to remove their brain and liver. The remaining rats (40) were grouped as follows: control group comprised 10 rats and were treated with distilled water. Experimental groups 1, 2, and 3 were of 10 rats each and were treated with sodium saccharine dissolved in water at 2.5, 5, and 10 mg/kg of body weight, respectively. The doses were chosen based on acceptable daily intake (ADI) (5mg/kg). All the treatments were given orally via gavage once a day over 120 days.

#### 2.2.3. Collection of Samples

Blood samples (3 ml) were collected in vivo from the orbital venous plexus [37] of the treated and control groups, 10 samples/group. The samples were collected at different time points, at time zero, and after 60 and 120 days of treatment. Blood samples were collected in a plain tube without anticoagulant and kept at room temperature for 30–60 min to clot. Sera were separated by centrifugation at 3000 rpm for 15 min. The obtained sera were used for the measurement of biochemical parameters. Urine samples were also collected at different time points, time zero and after 60 and 120 days to measure the concentration of isoprostane. Brain and liver tissues were surgically removed from five animals/group at at time zero, and after 120 days from the remaining animals. The collected tissues were immediately washed with ice-cold phosphate-buffered saline (PBS) (pH 7.4, 0.01 M). Further dissection was made on an ice-cold glass plate, and used for the assay of 8-OHdG and determination of catalase activity.

#### 2.2.4. Methods

To evaluate the adverse effects of saccharin on biochemical alterations in blood and tissues, enzymatic methods were applied to measure the serum level of glucose [38], TC [39], and TG [40]. The level of HDL was measured according to Burtis et al. (1999) [41]. LDL concentration was calculated according to the Friedewald formula [42], as follows:LDL-C (mg/dl) = TC- (HDL-C+ VLDL-C),(1)

Total protein concentration was determined by the Biuret method [43]. Serum albumin level was measured according to Doumas et al. (1971) using bromocresol green reagent [44]. 

Serum creatinine level was assayed according to Jaffe’s method using alkaline picrate reagent [45] and the level of uric acid was measured with the methods as described by Fosatti et al. (1971) [46]. The activity of AST and ALT were assayed according to the method of Reitman and Franke (1957) [47], and LDH activity was measured using the method of Youngs (1995) [48], whereas the activity of liver catalase was assayed according to Hadwan and Abeds (2016) [36]. 

Finally, for lipid peroxidation, and to detect DNA oxidation in response to toxicity of saccharine, isoprostane and 8-OHDG levels were determined by double-sandwich ELISA method described by Morrow et al. (1990) [49] and Souza-Pinto et al. (2001) [50].

### 2.3. Statistical Analysis

All data were analyzed by one-way analysis of variance (ANOVA). Specific differences between the groups were determined using the Duncan multiple range test [51]. The accepted level of significance was *p* < 0.05. Values were expressed as mean ± standard error of the mean (SEM) of rats (*n* = 10) per group.

## 3. Results

To determine the effect of the saccharin consumption on body weight and blood glucose, different doses of saccharin were given to the rat groups compared to the control. The observed data are shown in Figure 2a,b. A significant increase in body weight during the period of saccharin consumption was noticed at 5 mg/kg after 60 (532 ± 88.77 g, 59%) and 120 days (561 ± 16.18 g, 67%) of treatment, in comparison with the control group (335 ± 8.15 g). Saccharin consumption increased glucose level at 5 (75% and 62%) and 10 mg/kg (43% and 40%) after 60 and 120 days of treatment, respectively. The highest increase was observed after 60 days of treatment with 5 mg/kg (198.22 ± 6.55 mg/dl) and 10 mg/kg (171.31 ± 8.93 mg/dl), when compared with the control group (123.93 ± 6.25). 

The effects of saccharin consumption on lipid profile was detected and the data revealed that TC was decreased after 60 (21% and 22%) and 120 days (22% and 20%) when saccharin was given at 5 and 10 mg/kg, respectively, compared to the control group (Figure 3). Similarly, the concentration of TG was decreased significantly after 60 days of treatment with 2.5 mg/kg (54.91 ± 4.87 mg/dL, 37%) and 10 mg/kg (55.56 ± 3.53 mg/dL, 35%) when compared with the control group (73.29 ± 7.82 mg/dL). However, no significant variation was noticed in LDL levels, except after 60 days of saccharin treatment at 5 mg/kg (50.42 ± 3.66 mg/dL, 28%) and 10 mg/kg (43.7 ± 4.17 mg/dL, 23%) compared to the control group (69.49 ± 4.69 mg/dL). HDL concentration was decreased after treatment with saccharin at 2.5 mg/kg (48.76 ± 3.82 mg/dL, 23%) (46.26 ± 4.06 mg/dL, 27%) after 60 and 120 days, respectively, and 5 mg/kg (44.21 ± 3.18 mg/dL, 31%) after 120 days of treatment, when compared to the control group (66.12 ± 3.70 mg/dL).

The effect of saccharin intake at 2.5, 5 and 10 mg/kg from time zero to 60 and 120 days on total protein and albumin levels are shown in Figure 2c,d. The concentration of albumin was decreased after treatment with saccharin at 2.5 mg/kg (34% and 36%), 5 mg/kg (39% and 34%), and 10 mg/kg (15% and 21%) after 60 and 120 days of treatment, respectively. However, the concentration of total protein decreased (12%) after 120 days of treatment at all doses.

The effect of two different doses of saccharin on ALT in treated rats was detected and the observed data revealed a significant increase in the activity of ALT in the treated groups with saccharin at 5 mg/kg (62.42 ± 2.29 IU/L, 63%) and 10 mg/kg (73.53 ± 1.67 IU/L, 90%) after 60 days in comparison with the control group (39.3 ± 0.77 IU/L). The activity of ALT returned to normal levels after 120 days of treatment with saccharin at 2.5 and 5 mg/kg. However, after treatment with saccharin at 10 mg/kg, the activity of ALT increased (55.56 ± 56 IU/L, 43%) without returning to normal compared to the control group (41.32 ± 0.76 IU/L) (Figure 4b). The activity of AST increased proportionally to the consumption period (Figure 4a). The highest AST activity was observed at 120 days post-treatment at doses of 2.5 mg/kg (91.31 ± 6.16 IU/L, 19%) and 5 mg/kg (94.76 ± 3.14, 20%) compared to the control group (74.11 ± 2.22).

Serum creatinine, uric acid, LDH, and urinary isoprostane were also investigated in saccharin-treated groups, and the recorded data revealed a significant increase in the level of serum creatinine at 2.5 (125% and 68%), 5 (114% and 45%), and 10 mg/kg (26% and 31%) of saccharin-treated groups after 60 and after 120 days, respectively. The concentration of uric acid also increased after treatment with 2.5 (63% and 119%), 5 (132% and 106%), and 10 mg/kg (183% and 160%) after 60 and 120 days of treatment, respectively. LDH activity increased after treatment with saccharin at 2.5 (344% and 159%), 5 (373% and 263%), and 10 mg/kg (354% and 189%) after 60 and 120 days, respectively. Urinary isoprostane levels increased after treatment with saccharin at 2.5 (647% and 621%), 5 (539% and 1095%), and 10 mg/kg (620% and 675%) after 60 and 120 days, respectively. The uric acid and creatinine data are shown in Figure 4c,d, respectively. The LDH and isoprostane data are also shown in Figure 5c,d, respectively.

The activity of liver catalase tested in treated groups and the data revealed increases of liver catalase after 120 days of saccharin at 2.5 mg/kg (82.25 ± 2.47 kU/g, 51%) treatment compared to the control group (54.05 ± 1.73 kU/g) (Figure 5a). 

Finally, the recorded concentrations of 8-OHdG are shown in Figure 5b: 8-OHdG increased after 120 days of treatment with saccharin at 2.5 mg/kg (98.03 ± 5.12), 5 mg/kg (81.93 ± 3.14), and 10 mg/kg (81.85 ± 2.98) compared to the control group (65.43 ± 1.75). 

## 4. Discussion

Recently, concerns have been raised about the safety of artificial low-calorie sweeteners that are commonly used as substitutes for sucrose in many diet products. Studies have revealed that the use of artificial sweeteners may present some risks to users like migraines, headaches, skin eruptions, muscle dysfunction, depression, weight gain, liver and kidney toxicity, multiple sclerosis, blurred vision, respiratory problems, cancers, allergies, seizures, and immune system dysfunction [52,53].

Little was known about the toxicity of the artificial sweeteners, therefore, the present study was designed to give a better understanding about the toxicity of saccharin, particularly the mechanisms leading to dyslipidemia, glucose homeostasis impairment, oxidative stress, hepatotoxicity, and kidney dysfunction. The adverse effects of saccharin on biochemical alterations in blood and tissues of rats treated with different doses of sodium saccharine were analyzed for 120 days.

The recorded data found a correlation between the consumption of sodium saccharin and the risk of obesity; administration of 5 mg/kg and 10 mg/kg of sodium saccharin greatly increased body weight after 60 and 120 days. It is believed that the increases within the body weight were due to glucose intolerance in treated groups, where artificial sweeteners including saccharin, sucralose, and aspartame were found to induce microbial communities in the digestive system [54]. 

Sweetness, when not accompanied by calories, results in ambiguous psychobiological signals that bewilder the body’s regulatory mechanisms, leading to a loss of control over appetite and overeating [55,56]. As a result, intense sweeteners have been blamed for the obesity epidemic [56,57]. A study conducted on animals for type 2 diabetes showed that the consumption of artificial sweeteners can influence diabetes-related parameters and complications associated with diabetes [58]. The results corroborate the hypothesis that the consumption of artificial non-nutritive sweeteners is linked to obesity or obesity-like effects [59]. 

Swithers and Davidson in their study stated that sweet tastes normally indicate that the body is about to receive a high number of calories, and the digestive system prepares to react accordingly. When sweet tastes are not followed by the expected number of calories, which happens with sweeteners including saccharin, the body becomes conditioned against the expected response to the calories received [60]. 

The present study recorded increases in blood glucose that were observed in dose-dependent saccharin-treated groups. These data are in line with those described by Prokić et al. (2015). Moreover, it has been reported that sweeteners cause a cephalic phase insulin release that decreases blood glucose levels [4,25,61,62,63,64]. A mechanism suggested by Swithers et al. (2013) hypothesizes that exposure to high-intensity sweeteners (20% glucose, 0.3 % saccharine) interferes with a predictive relationship between sweet tastes and calories that may impair energy balance, which could in turn alter glucose homeostasis and reduce satiety [65].

The effects of saccharin consumption on lipid profile were also evaluated in this study. The concentrations of TC, TG, and LDL- and HDL-cholesterol were decreased in response to oral treatment with sodium saccharin and were observed after 60 and 120 days of treatment compared to the control group. These findings reflect the conclusion found in previous studies [4,64,66].

The reductions within the lipid profile could be due to the indirect action of saccharin on the lipid metabolism and on lipid peroxidation, which has been previously observed [67]. It has been demonstrated that compounds presenting cyclic imides, including saccharin, can interfere with lipoprotein receptor binding and degradation of rat and human cells while affecting regulatory enzymes of lipid metabolism [68]. 

The hypocholesterolemic and hypolipidemic effect found in the present study could perhaps be attributed to a reduced total cholesterol biosynthesis. Since it is known that saccharin can suppress liver enzymatic activity in vivo, the activity of acetyl-CoA synthetase and citrate lyase could be reduced by the consumption of saccharin. Besides the concentration of mitochondrial citrate, these effects would lead to a reduction of available cytoplasmic acetyl-CoA, a precursor for the synthesis of cholesterol and fatty acids [69,70]. Moreover, liver acetyl-CoA carboxylase, phosphatidate phosphohydratase, and glycerol-3-phosphate acyl transferase activities were shown to be markedly reduced by saccharin analogues [69]. Suppression of those enzymes would result in a reduction of TG biosynthesis [25]. Another mechanism that could be contributing to the decreased concentration of lipid profile parameters would be related to ApoA. ApoA is the major protein constituent of HDL, and long-term consumption of artificial sweeteners may cause modification of this protein, which is associated with adverse effects on antioxidizing processes and impairment of the phospholipid binding ability of HDL [71]. 

The effect of oral consumpation of saccharin at 5 mg/kg and 10 mg/kg on ALT activity was detected and the data found ALT activity increased after 60 days and 120 days of treatment. These findings are in line with [4,9] who reported that a low dose of 10 mg/kg.b.w. and a high dose of 500 mg/kg.b.w. of saccharin exhibited a significant increase in the activity of ALT, AST, ALP, and serum markers of liver function. This alteration was suggested as a common sign of impaired liver function [4,9]. The elevation in serum aminotransferase activity could be due to an effect caused by free radical interaction with cellular membranes or could be related to breakdown of liver parenchyma [72]. The changes in liver function could be attributed to a hepatocellular impairment. Subsequently, this alteration would cause the release of abnormal levels of intracellular enzymes into the blood. The elevation in the activity of aminotransferase indicates an early diagnosis of hepatotoxicity and is considered a biomarker of tissue damage [29] 

LDH activity when compared to other liver function tests (LFTs) stayed elevated at all doses and during the study period, including the acceptable daily intake (ADI). Recently, similar results were obtained when using aspartame as a sweetener instead of saccharin [73]. Interestingly, the above effects of saccharin on LFTs paralleled the results obtained on total protein and albumin levels. As it was mentioned previously, the administration of sodium saccharin led to liver dysfunction through its effect on liver enzymes. Therefore, the levels of total protein and albumin could also be affected as a result of hepatocellular injury and dysfunction [74,75]. Another possibility would be the decreased blood protein concentration as a consequence of decreased protein synthesis or increased proteolytic activity [76]. 

Chronic consumption of saccharin may cause kidney injury, according to the results, which showed a significant elevation in the blood creatinine level at all doses studied when compared to the control group. This result could be due to disturbances in renal functions leading to reduced glomerular filtration rate followed by retention of urea and creatinine in the blood [77]. Similar results were obtained previously showing that saccharin at doses between 10 and 500mg/kg may harmfully alter biochemical markers in the liver and kidneys [9,29,75]. 

ROS, including free radicals, H_2_O_2_, and peroxides that could lead to cell damage are produced at a low level by normal aerobic metabolism so that the damage to cells is constantly repaired. However, severe levels of oxidative stress causes ATP depletion, preventing controlled apoptotic cell death which can lead to DNA damage. The oxidative stress induced by high doses of saccharin could be attributed to the inflammation of the liver cells [29]. The inflammatory process induces oxidative or nitrosative stress and lipid peroxidation, thereby generating excess ROS and RON (superoxide anion and H_2_O_2_), in addition to secondary oxidants and DNA-reactive aldehydes [78]. Increased oxidative stress in saccharin-treated groups may not be due to lipid profile elevation but instead a result of impaired liver function [29].

Data obtained using aspartame showed that oxidizing conditions can lead to increased levels of 8-OHdG, a product of deoxyguanosine hydroxylation in DNA [79]. Therefore, 8-OHdG has been proposed as an indicator of oxidative damage in DNA in vivo and in vitro [80,81]. Arachidonic acid, under certain conditions, undergoes auto-oxidation resulting in the formation of prostagladin-like compounds such as isoprostane [49]. As a result, 8-iso-PGF2 has been considered a marker of oxidative stress under clinical and experimental conditions [82,83]. Similarly, the present study also observed an increased concentration of 8-OHdG in all studied groups who received the treatment for 120 days, particularly in groups receiving the lower dose of saccharin. In addition, the urinary isoprostane increased proportionally to treatment time and dosage size. These findings were corroborated by observed increases in catalase activity, especially in rats treated with saccharin at 2.5 mg/kg. Our findings represent the first attempt to measure the effects of long-term saccharin consumption on 8-OHdG and isoprostane.

A positive relationship between oxidative stress and saccharin consumption was found. The lowest dose of saccharin can lead to a release of ROS which could be overcome by the body’s ability to produce antioxidants, such as catalase, as a result of cellular protection against ROS. This mechanism of homeostasis involves an elaborate antioxidant defense system, including several antioxidant enzymes [84,85]. If the consumption of saccharin persists and the dose increases free radical production, it could potentially overwhelm antioxidant mechanisms and lead to a decrease in catalase activity. 

Purines can be released through tissue injury and hypoxia. It is known that hypoxia is a potent inducer of xanthine oxidase. This enzyme uses molecular oxygen in place of NAD^+^ as an electron acceptor to induce the formation of superoxide anions and hydrogen peroxide in parallel with uric acid [86,87]. The concentration of uric acid increased proportionally to the dose and period of saccharine consumption. The highest uric acid concentration was obtained following 120 days of saccharin consumption at its highest dose (10 mg/kg). The results obtained with the uric acid can also be understood as a response to the oxidative stress induction caused by saccharin consumption. Even though uric acid has antioxidant properties, under some conditions its antioxidant capacity can be overcome by pro-oxidant and proinflammatory effects of ROS accumulation [88].

## 5. Conclusions

The results obtained in the present study suggest that long-term saccharin consumption increases the risk of obesity and diabetes, as well as liver and renal impairment. The results also suggest an increased risk of brain carcinogenesis. 

## Figures and Tables

**Figure 1 medicina-55-00681-f001:**
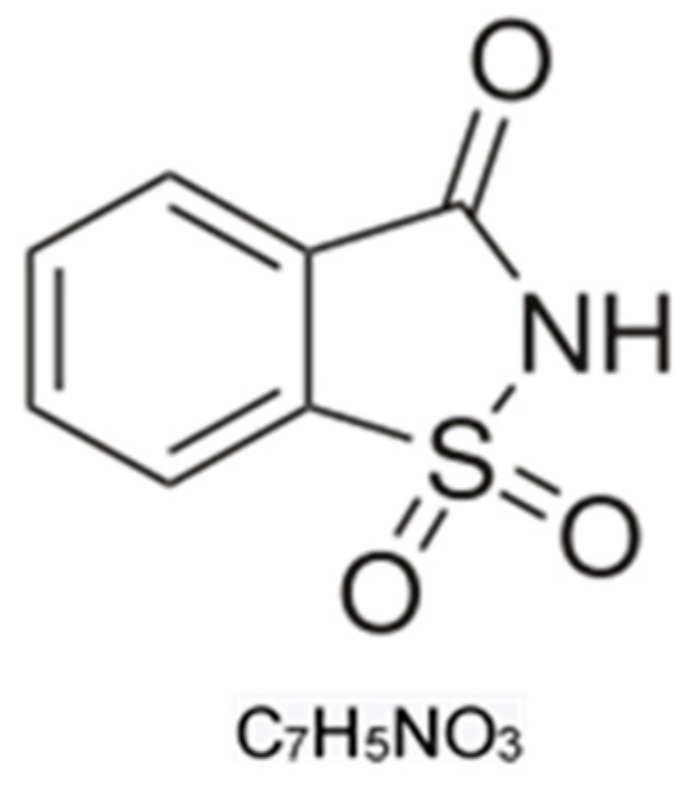
Saccharin (1,2-benzisothiazol-3(2H)-one-1,1-dioxide) chemical structure and molecular formula.

**Figure 2 medicina-55-00681-f002:**
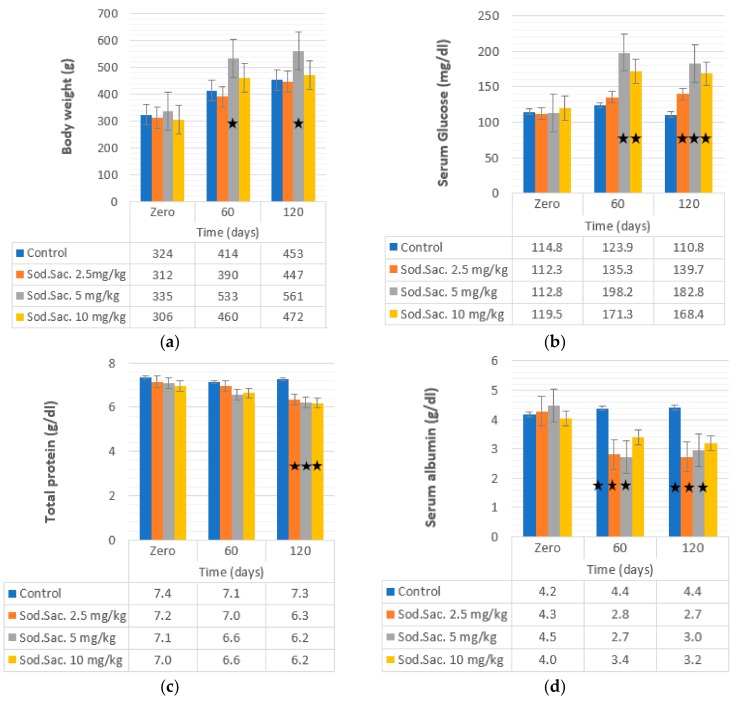
Effect of the administration of sodium saccharine in rats (*n* = 10) at 2.5, 5, and 10 mg/kg.b.w. at 0 time and after 60 and 120 days on (**a**) body weight, (**b**) glucose, (**c**) TP, and (**d**) albumin. Comparison between the groups was made by the one-way analysis of variance (ANOVA) with mean ± SEM. Significant (*) at *p* < 0.05.

**Figure 3 medicina-55-00681-f003:**
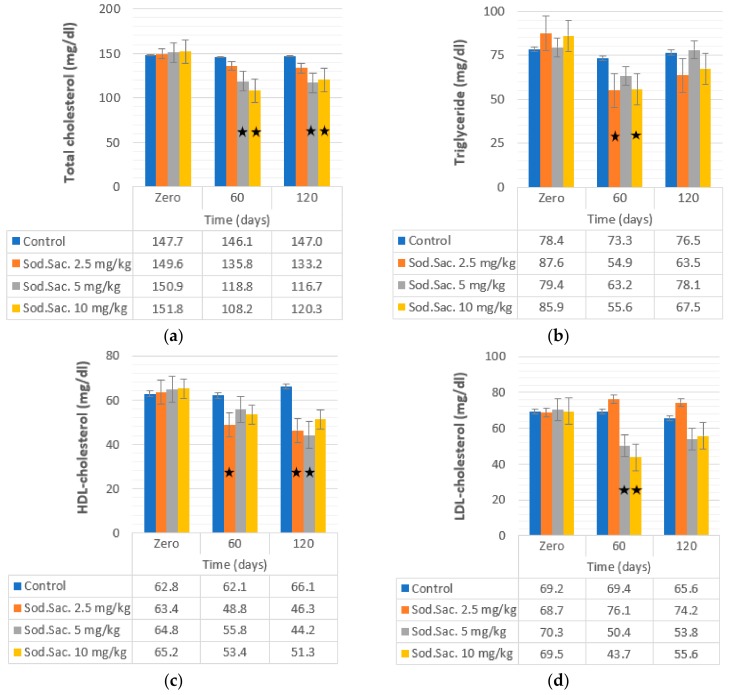
Effect of the administration of sodium saccharine in rats (*n* = 10) at 2.5, 5, and 10 mg/kg.b.w. at 0 time and after 60 and 120 days on (**a**) total cholesterol, (**b**) triglyceride, (**c**) HDL-cholesterol, and (**d**) LDL-cholesterol. Comparison between the groups was made by one-way analysis of variance (ANOVA) with mean ± SEM. Significant (*) at *p* < 0.05.

**Figure 4 medicina-55-00681-f004:**
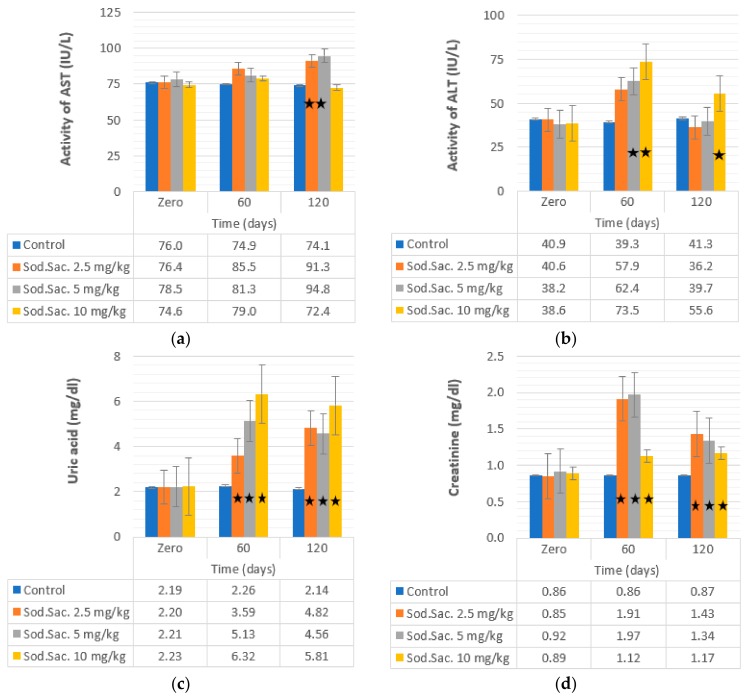
Effect of the administration of sodium saccharine in rats (*n* = 10) at 2.5, 5, and 10 mg/kg.b.w. at 0 time and after 60 and 120 days on (**a**) AST, (**b**) ALT, (**c**) uric acid, and (**d**) creatinine. Comparison between the groups was made by the one-way analysis of variance (ANOVA) with mean ± SEM. Significant (*) at *p* < 0.05.

**Figure 5 medicina-55-00681-f005:**
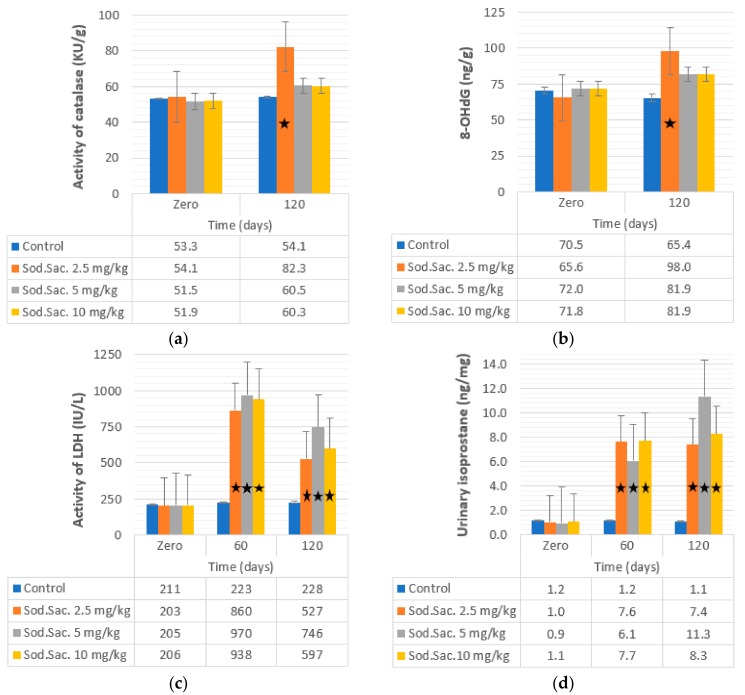
Effect of administration of sodium saccharine in rats (*n* = 10) at 2.5, 5, and 10 mg/kg.b.w. at 0 time, and after 60 and 120 days on (**a**) catalase, (**b**) 8-OHdG, (**c**) LDH, and (**d**) urinary isoprostane. Comparison between the groups was made by the one-way analysis of variance (ANOVA) with mean ± SEM. Significant (*) at *p* < 0.05.

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
