# Peer review of "Long-Term Saccharin Consumption and Increased Risk of Obesity, Diabetes, Hepatic Dysfunction, and Renal Impairment in Rats"

_medicina, 2019, doi:10.3390/medicina55100681_

Round 1
Reviewer 1 Report
After making corrections by the authors, I have no more comments on
the text of the manuscript.
Author Response
Thank you for your comments, and I worked to improve the language of the article. I send to you the final version of the article
Regards
corresponding Author
Omar Hasan Azeez
Reviewer 2 Report
Interesting study of an interesting topic. You treated animals during a long period and it is OK to interpret results as chronic effects of the saccharine. Methodology is valid and it could be repeated if needed. this paper should be published after some minor language editing.
Author Response
Thank you for your comments, and I improved the language of the article. I send to you the final version of the article.
Regards
Corresponding Author
Omar Hasan Azeez
Reviewer 3 Report
Manuscript is well written and organized. It can be acepted after authors made minor changes
Authors should add administration route to material and method. Gavage, ip or... Please edit "form" as "from" (page 4 line 150) Please check whole manuscript for references. There is some mistakes
Author Response
Thank you for your comments. I worked on improving the language of the whole article, specially the methods. You can check the changes from the last version of the article that I send to you.
Regards
Corresponding Author
Omar Hasan Azeez
This manuscript is a resubmission of an earlier submission. The following is a list of the peer review reports and author responses from that submission.
Round 1
Reviewer 1 Report
The manuscript has some editioral, punctuation errors - it should be corrected.
English should be improved.
Experimental Design - this part is unclear for the reviewer.
How many animals were at the very beginning of the experiment? 7-10 +5?; It is very unclear!!!
And what does it mean that n=7-10? The number of animals in each group should be the same.
The end point is unclear too. If 5 animals were taken for the assay of 8-OHdG and determination of catalase activity so that n=5 (for statistical analysis).
And what about the amount of the blood for analysis? At the end point was the blood taken in vivo?
Reviewer 2 Report
Page2, line 70-71: these two sentences are conflicting - first you say that it raises insulin levels in rats, and after that it is excreted and it does not affect insulin. So, you should either say that there is not enough evidence or explain if it is related to different species?!
In general it is very nice experiment, covering long period of 120 days. The analyses that you performed are showing both metabolic and oxidative stress parameters and mechanisms. Results that you have presented in clear and understandable way, are interesting and show no contradictions with each other.
Reviewer 3 Report
I have reviewed the manuscript entitled "Long-Term Saccharin Consumption and Increased Risk of Obesity, Diabetes, Hepatic Disfunction and Renal Impairment in Rats". The manuscript deals with toxicity of the Saccharin. It can be accepted after authors made some changes. My comments are below
How the authors selected doses? What was the criteria? Authors should add some information from "Yilmaz S and Uçar A. Saccharin genotoxicity and carcinogenicity: a review" to "Saccharin and toxicity" section in Introduction.

Reviewer 4 Report
Overall, a significant study is submitted. Experimental design is appropiate. All figures are informative and well presented. English is acceptable. Discussion is deep and comparative. Many sources are cited.
Technical error: on figure 4D "Ceratinine" should be replaced by "Creatinine".
----------------
The manuscript was withdrawn by author's request after receiving the reports. Responses below were provided on the new submission cover letter. In house editors contacted same previous reviewers; reviewer 1 = reviewer 1, ...
Author Response reviewer 1 report
a-The experimental design was rewritten and better explained.
b- The number of animals used to each experiment was specified. The modifications can be seen in the lines 144-150.
“Rats were divided into four groups (15 rats/group). For the experiment base line (time zero) five rats from each group were euthanized to remove their brain and liver. The remaining rats (40) were divided as: Control group (10 rats): received distilled water. Experimental Groups 1, 2, and 3 (10 rats/group): treated with increasing doses of sodium saccharine dissolved in water at 2.5, 5 and 10 mg/kg (body weight), respectively. The doses were chosen based on ADI (5mg/kg). All the treatments were given orally via gavage once a day during 120 days.”
- c) Regarding to the assay of catalase activity and 8-OHdG concentration, the text was revised and rewritten accordingly. All the blood samples were taken in vivo as now was added to the text. The modifications can be seen in the lines 152-161.
“Blood samples (3 ml) were collected in vivo from the orbital venous plexus ]38[ form the rats (10 sample/group) at time zero, and after 60 and 120 days. These samples were collected into a glass tube without anticoagulant allowing it to clot at room temperature for 30-60 min. Serum was separated by centrifugation at 3000 rpm for 15 min. The serum was used for the measurement of biochemical parameters. Urine samples were collected at time zero and after 60 and 120 days to measure the concentration of isoprostane. Finally, samples of brain and liver tissues were removed by surgery at time zero from 5 animals selected from each group, and after 120 days from the remaining animals. The tissues were immediately washed with ice-cold phosphate buffered saline (PBS) (pH7.4, 0.01M). Further dissection was made on ice-cold glass plate, and used for the assay of 8-OHdG and determination of catalase activity.”
Author Response reviewer 2 report
The text mentioned by the reviewer was revised and rewritten accordingly. The apparently conflicting results mentioned in the text were possibly due to the application of two different animal models and period of treatment with the sweetener, as explained in this new text version. The modifications can be seen in the lines 74-79.
“Since saccharin is not metabolized and doesn’t produces food energy it became an important sweetener especially for diabetics [14]. Since the discover of saccharin its effect on insulin release has been controversial. According to Ionescu et al (1988) saccharin could trigger the release of insulin in genetically modified obese rats, most likely as a result of its taste [14]. However, Whitehouse et al (2008) showed that saccharin doesn’t affect blood insulin level when the animals are treated with the sweetener from the time of conception to death [15].”
Author Response reviewer 3 report
a- The doses of sodium saccharin were chosen according to the acceptable daily intake (ADI: 5 mg /kg, body weight). According to the experimental design proposed for the present research a dose inferior (2.5 mg/kg) and a dose superior (10 mg/kg) to the ADI were also included. The criteria to choose the dose was added to the new text version and can be seen in the line 149.
b- The reference suggested by the reviewer (Uçar, A.; Yilmaz, 2015 Yilmaz and Uçar, 2015) was included. This modification can be seen in the lines 91-92.
Author Response reviewer 4 report
The term Creatinine was corrected in figure 4D